# VALUE-ANCHORED GROUP POLICY OPTIMIZATION FOR FLOW MODELS

## ABSTRACT

Group Relative Policy Optimization (GRPO) has proven highly effective in enhancing the alignment capabilities of Large Language Models (LLMs). However, current adaptations of GRPO for the flow matching-based image generation neglect a foundational conflict between its core principles and the distinct dynamics of the visual synthesis process. This mismatch leads to two key limitations: (i) Uniformly applying a sparse terminal reward across all timesteps impairs temporal credit assignment, ignoring the differing criticality of generation phases from early structure formation to late-stage tuning. (ii) Exclusive reliance on relative, intra-group rewards causes the optimization signal to fade as training converges, leading to the optimization stagnation when reward diversity is entirely depleted. To address these limitations, we propose Value-Anchored Group Policy Optimization (VGPO), a framework that redefines value estimation across both temporal and group dimensions. Specifically, VGPO transforms the sparse terminal reward into dense, process-aware value estimates, enabling precise credit assignment by modeling the expected cumulative reward at each generative stage. Furthermore, VGPO replaces standard group normalization with a novel process enhanced by absolute values to maintain a stable optimization signal even as reward diversity declines. Extensive experiments on three benchmarks demonstrate that VGPO achieves state-of-the-art image quality while simultaneously improving task-specific accuracy, effectively mitigating reward hacking. The code will be made available to the public.

## 1 INTRODUCTION

The evolution of aligning Large Language Models (LLMs) (Guo et al., 2025; Jaech et al., 2024) with human intent has recently entered a new phase (Ouyang et al., 2022). Initially dominated by supervised instruction tuning, the field is now increasingly leveraging the principles of reinforcement learning (RL) (Kaelbling et al., 1996) to achieve more sophisticated behavioral control. This transition is marked by recent advances such as PPO (Schulman et al., 2017), DPO (Rafailov et al., 2023) and GRPO (Shao et al., 2024), which represent a departure from training on static datasets toward interaction-based optimization of policies driven by human preference signals.

Reinforcement learning for flow-based generative models remains comparatively underexplored, in contrast to diffusion-based generative models (Black et al., 2023; Wallace et al., 2024). Most efforts (Liu et al., 2025b; Chen et al., 2025a) directly transplant paradigms developed for large language models into the flow matching setting with minimal adaptation, neglecting the distinctive characteristics of generative process dynamics. For instance, Flow-GRPO (Liu et al., 2025a) and DanceGRPO (Xue et al., 2025) propose to directly apply the advanced GRPO (Shao et al., 2024) algorithm to state-of-the-art text-to-image flow matching models (Esser et al., 2024; Labs, 2024) through exploring the action space using SDE sampling methods (Song et al., 2020).

However, these methods tend to overlook the potential mismatch between the assumptions of GRPO and the dynamics of the flow matching environment. First, GRPO assumes that the action values are uniform across all intermediate steps. In the environment of flow matching models, the progressive transformation of Gaussian noise into a high-quality image introduces intermediate actions of varying values. By distributing a uniform, terminal reward across all denoising steps, these methods ignore the differential impact of each step in the generation process, failing to distinguish between

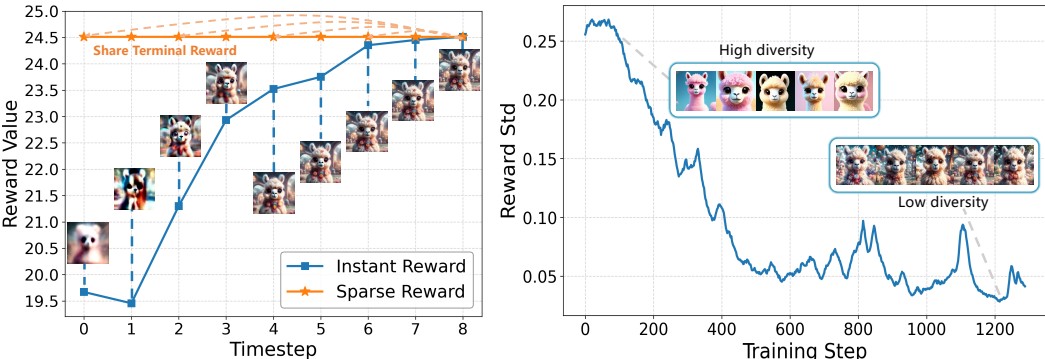

Figure 1: **Motivation.** (Left) **Sparse vs. Instant Reward Signals During Generation.** The sparse terminal reward remains constant, failing to provide varying values for intermediate steps. (Right) **Diminishing Reward Std as Policy Converges.** Due to the reliance on reward diversity, the reward std declines as the training process advances, potentially leading to optimization stagnation.

critical early stages for structure formation and later stages for fine-grained detail refinement. This indiscriminate credit allocation results in misleading optimization signals that impair sample efficiency and hinder effective learning (Yang et al., 2024b; Qu et al., 2025; Cui et al., 2025b). As shown in Fig. 1(Left), this sequential process enables the evaluation of intermediate action values, offering a more granular assessment than previous methods reliant solely on a share terminal reward. Second, GRPO (Shao et al., 2024) and its related methods (Yu et al., 2025) fundamentally leverage the diversity within final rewards to guide their optimization process. The optimization signal is only derived from the relative advantage, which depends on reward variance within a group. Nevertheless, our empirical result (Fig. 1(Right)) indicates that the diversity, which serves as the driving force for optimization, progressively diminishes as the optimization process advances. This can cause the optimization process to stagnate, when the model exclusively generates uniformly low/high-reward results within a rollout group. This vulnerability to stagnation is particularly acute in visual generation tasks compared to large language models, as models can more easily converge to a single aesthetic or stylistic mode (Lee et al., 2023).

To address these limitations, we propose **V**alue-**A**nchored **G**roup **P**olicy **O**ptimization (**VGPO**) framework built on two key components. First, we introduce the Temporal Cumulative Reward Mechanism (TCRM), which leverages Monte-Carlo estimation over the sampling trajectory to assess the value of intermediate actions. Specifically, we introduce the definition of an instant reward for a given state-action pair $(s_t, a_t)$, which subsequently allows to approximate the ground-truth intermediate action value using the available sampled trajectories. Second, to counteract the adverse effects of diminishing reward diversity, we propose the Adaptive Dual Advantage Estimation (ADAE). This replaces standard normalization with a novel process enhanced by absolute metrics for advantage computation. Critically, we can prove that ADAE automatically switches to optimizing absolute values when reward diversity is fully depleted. Extensive experiments on compositional image generation, visual text rendering and human preference alignment tasks demonstrate that VGPO enhances task-specific accuracy while significantly improves image quality and diversity compared to existing flow-based RL methods.

The contributions of this paper are as follows:

- We identify a fundamental mismatch between the core assumptions of GRPO and the dynamics of flow-based generation, leading to two critical limitations: misalignment between process exploration and outcome reward, and reliance on reward diversity.

- We propose VGPO, a framework built upon the synergistic action of the temporal cumulative reward mechanism for process-aware value estimation and the adaptive dual advantage estimation for stable advantage computation.

- We conduct comprehensive experiments on three benchmarks, demonstrating that VGPO achieves state-of-the-art performance by attaining higher alignment accuracy, promoting more efficient exploration, and mitigating reward hacking.

## 2 RELATED WORK

**RL for Diffusion Models.** Due to the significant effectiveness of RL in enhancing the reasoning capabilities of large language models (LLMs) (Jaech et al., 2024; Shao et al., 2024), its application to diffusion models has become a rapidly developing research direction. Early works (Yang et al., 2024a; Black et al., 2023; Fan et al., 2023; Lee et al., 2023) draw inspiration from classical policy gradient algorithms like Proximal Policy Optimization (PPO) (Schulman et al., 2017) to align pre-trained T2I models with human preferences. Subsequently, Diffusion-DPO (Wallace et al., 2024) and its variants (Liang et al., 2024; Dong et al., 2023; Yuan et al., 2024b;a) integrate Direct Preference Optimization (DPO) (Rafailov et al., 2023) into T2I generation to enable direct learning from preference data. Recent works (Li et al., 2025a; He et al., 2025; Li et al., 2025b; Wang & Yu, 2025; Wang et al., 2025a; Shen et al., 2025; Zheng et al., 2025) begin to explore the potential of online RL in advancing flow matching generative models. In particular, Flow-GRPO (Liu et al., 2025a) and DanceGRPO (Xue et al., 2025) are the first to incorporate advanced Group Relative Policy Optimization (GRPO) (Shao et al., 2024) into flow-matching models by converting ODE sampling into equivalent SDE. However, directly transplanting GRPO into the flow-matching setting fails to account for the mismatch between the algorithm's core assumptions and the intrinsic dynamics of flow matching. Our VGPO addresses this by introducing the Temporal Cumulative Reward Mechanism (TCRM) to establish a process-aware reward structure and the Adaptive Dual Advantage Estimation (ADAE) to prevent policy collapse by maintaining a stable optimization signal.

**Dense Process Rewards.** The challenge of credit assignment with sparse terminal rewards has driven the adoption of dense process rewards, which have proven effective in areas such as the inference-time scaling of LLMs (Lightman et al., 2023; Uesato et al., 2022; Cui et al., 2025a). TPO (Liao et al., 2024) extracts more fine-grained process rewards by ranking entire reasoning trajectories and adaptively assigning credit to the critical intermediate steps. Recent efforts to apply dense process rewards in diffusion models have explored two main paradigms. The first involves building explicit process reward models (PRMs), such as in SPO (Liang et al., 2024), which trains a model to evaluate intermediate steps. However, this method is often hampered by high annotation costs and the challenge of training on noisy images. The second paradigm infers process signals from terminal rewards. For example, DenseReward (Yang et al., 2024b) breaks temporal symmetry in DPO-style objectives by introducing temporal discounting. However, prior credit assignment methods are highly sample-inefficient, requiring full trajectory rollouts for single-step evaluation, and myopic, attributing terminal rewards to single actions without considering long-term value. Our VGPO resolves these limitations by efficiently estimating long-term cumulative values from one-step ODE sampling and in turn using them to re-weight timesteps, assigning greater importance to critical decisions in the generation process.

## 3 METHOD

### 3.1 PRELIMINARIES

**Flow Matching.** The Rectified Flow (Liu et al., 2022; Lipman et al., 2022) framework has emerged as a foundational technique for generative modeling, underpinning recent advances in both image and video generation. Central to this framework is the construction of a linear trajectory that connects a data sample $\boldsymbol{x}_0 \sim X_0$ with a noise sample $\boldsymbol{x}_1 \sim X_1$. A noisy latent $\boldsymbol{x}_t$ is defined as:

$$\boldsymbol{x}_t = (1-t)\boldsymbol{x}_0 + t\boldsymbol{x}_1 \tag{1}$$

By training the model to predict the velocity $\boldsymbol{v}$, the Flow Matching objective can be formulated as:

$$\boldsymbol{L}(\theta) = \mathbb{E}_{t,\boldsymbol{x}_0,\boldsymbol{x}_1} \|\boldsymbol{v} - \boldsymbol{v}_\theta(\boldsymbol{x}_t, t)\|^2 \tag{2}$$

where the target velocity field is $\boldsymbol{v} = \boldsymbol{x}_1 - \boldsymbol{x}_0$.

**GRPO.** Group Relative Policy Optimization (GRPO) (Shao et al., 2024) is a reinforcement learning method that leverages the average reward of multiple sampled outputs as a dynamic baseline for advantage estimation. This principle was recently adapted for generative models in Flow-GRPO (Liu et al., 2025a), which applies GRPO to improve the performance of state-of-the-art flow matching models (Wan et al., 2025; Esser et al., 2024; Labs, 2024). The underlying framework for this

approach, following prior work on diffusion models (Black et al., 2023), is to cast the iterative generation process as a Markov Decision Process (MDP), formulated as:

$$s_t \triangleq (c, t, x_t) \quad \pi(a_t \mid s_t) \triangleq p_\theta(x_{t-1} \mid x_t, c) \quad P(s_{t+1} \mid s_t, a_t) \triangleq (\delta_c, \delta_{t-1}, \delta_{x_{t-1}})$$
$$a_t \triangleq x_{t-1} \quad \rho_0(s_0) \triangleq (p(c), \delta_T, \mathcal{N}(\mathbf{0}, \mathbf{I})) \quad R(s_t, a_t) \triangleq \begin{cases} r(x_0, c) & \text{if } t = 0 \\ 0 & \text{otherwise} \end{cases} \quad (3)$$

where at each timestep $t$, the agent observes a state $s_t$, takes an action $a_t$, receives a reward $R(s_t, a_t)$, and transitions to a new state $s_{t+1} \sim P(s_{t+1} \mid s_t, a_t)$. The agent acts according to a policy $\pi(a_t \mid s_t)$ and $\rho_0(s_0)$ represents the initial-state distribution.

Given a prompt $c$, the flow model $p_\theta$ samples a group of $G$ individual images $\{x_0^i\}_{i=1}^G$ and the corresponding reverse-time trajectories $\{x_T^i, x_{T-1}^i, \cdots, x_0^i\}_{i=1}^G$. Then, the advantage of the $i$-th image is calculated by normalizing the group-level rewards as follows:

$$\hat{A}_t^i = \frac{R(x_0^i, c) - \text{mean}\left(\{R(x_0^i, c)\}_{i=1}^G\right)}{\text{std}\left(\{R(x_0^i, c)\}_{i=1}^G\right)} \quad (4)$$

Flow-GRPO optimizes the policy model by maximizing the following objective:

$$\mathcal{J}_{\text{Flow-GRPO}}(\theta) = \mathbb{E}_{c \sim \mathcal{C}, \{x^i\}_{i=1}^G \sim \pi_{\theta_{\text{old}}}(\cdot \mid c)}$$
$$\left[\frac{1}{G} \sum_{i=1}^G \frac{1}{T} \sum_{t=0}^{T-1} \left(\min\left(r_t^i(\theta)\hat{A}_t^i, \text{clip}\left(r_t^i(\theta), 1-\varepsilon, 1+\varepsilon\right)\hat{A}_t^i\right) - \beta D_{\text{KL}}(\pi_\theta \| \pi_{\text{ref}})\right)\right] \quad (5)$$

where

$$r_t^i(\theta) = \frac{p_\theta(x_{t-1}^i \mid x_t^i, c)}{p_{\theta_{\text{old}}}(x_{t-1}^i \mid x_t^i, c)} \quad (6)$$

Flow matching models which utilize deterministic ODE-based sampling, inherently lack the stochasticity required for the probabilistic policy updates in GRPO. To address this, Flow-GRPO converts the deterministic ODE into an equivalent SDE that matches the original model's marginal probability density function at all timesteps (Song et al., 2020; Albergo et al., 2023; Domingo-Enrich et al., 2024). The final update rule is formulated as:

$$x_{t+\Delta t} = x_t + \left[v_\theta(x_t, t) + \frac{\sigma_t^2}{2t}(x_t + (1-t)v_\theta(x_t, t))\right]\Delta t + \sigma_t\sqrt{\Delta t}\epsilon \quad (7)$$

where $\sigma_t = a\sqrt{\frac{t}{1-t}}$ and $a$ is a scalar hyper-parameter that controls the noise level.

## 3.2 MOTIVATIONS

### 3.2.1 MISALIGNMENT BETWEEN PROCESS EXPLORATION AND OUTCOME REWARD

The core limitation of applying GRPO to flow matching models is the temporal misalignment inherent in its objective function. This issue stems from coupling a time-dependent policy ratio, which represents the process exploration, with a time-independent outcome reward, formulated as:

$$\mathcal{J} = \mathbb{E}\left[\underbrace{\frac{p_\theta(x_{t-1}^i \mid x_t^i, c)}{p_{\theta_{\text{old}}}(x_{t-1}^i \mid x_t^i, c)}}_{\text{process exploration}} \cdot \underbrace{A(x_0)}_{\text{outcome reward}}\right] \quad (8)$$

where the stepwise advantage remains constant by setting $A_t \equiv A(x_0)$ according to the final result. This formulation effectively distributes a uniform, sparse terminal reward across all timesteps, ignoring the differential impact of each action in the generative sequence. As illustrated in Fig. 1(Left), such indiscriminate credit assignment fails to capture the true value evolution as an image is progressively refined from noise. For instance, it may unduly penalize critical early-stage structural decisions or reward trivial late-stage refinements, resulting in misleading optimization signals that impair learning efficiency (Guo et al., 2021). To resolve this misalignment, we propose a forward-looking temporal cumulative reward mechanism that aligns the reward signal with the exploration process, enabling more precise and efficient policy optimization.

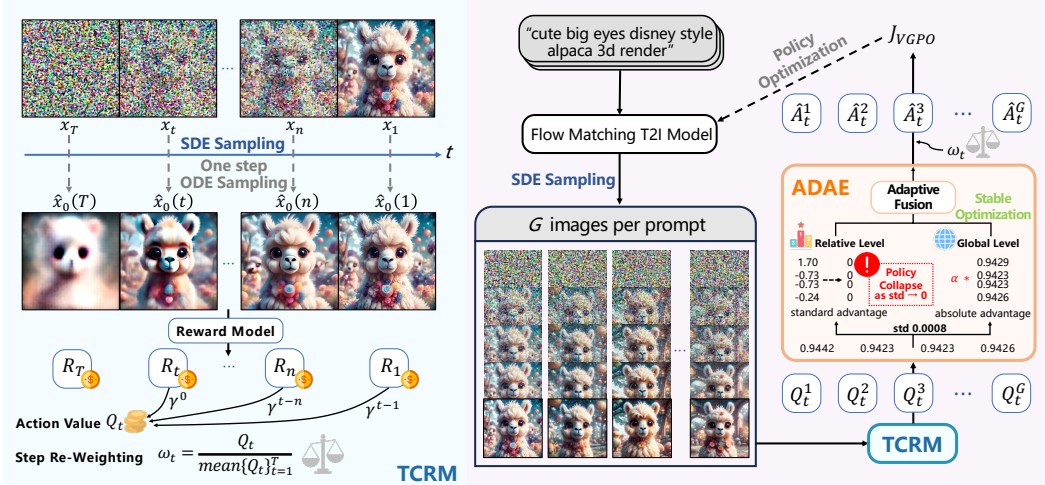

Figure 2: **Method Overview.** First, to resolve faulty credit assignment, Temporal Cumulative Reward Mechanism (TCRM) transforms sparse terminal rewards into dense, forward-looking process values, enabling a more granular, temporally-aware credit assignment. Second, to counteract policy collapse, Adaptive Dual Advantage Estimation (ADAE) replaces standard normalization with a novel process enhanced by absolute values for advantage computation, ensuring a persistent optimization signal that remains stable even when reward diversity diminishes.

### 3.2.2 RELIANCE ON REWARD DIVERSITY

The GRPO framework leverages reward diversity within each generated group to derive its optimization signal, as its advantage function is normalized by the reward standard deviation (Eq. 4). This driving force of optimization progressively diminishes as the optimization process advances, as shown in Fig. 1(Right). This dependency leads to policy stagnation whenever reward diversity is depleted, regardless of the absolute quality of the samples. To address it, we propose adaptive dual advantage estimation, which ensures a persistent optimization gradient by decoupling the learning signal from reward variance, enabling continuous exploration towards higher-quality outcomes.

### 3.3 VALUE-ANCHORED GROUP POLICY OPTIMIZATION

In this section, we introduce Value-Anchored Group Policy Optimization (VGPO), a novel framework (Fig. 2) that makes two core contributions: Temporal Cumulative Reward Mechanism (Sec. 3.3.1) and Adaptive Dual Advantage Estimation (Sec. 3.3.2).

### 3.3.1 TEMPORAL CUMULATIVE REWARD MECHANISM

We introduce the Temporal Cumulative Reward Mechanism (TCRM) to transform sparse terminal rewards into a dense, forward-looking value signals for precise credit assignment. First, we define an instant reward for each state-action pair $(s_t, a_t)$, which subsequently enables the approximation of ground-truth intermediate action values using sampled trajectories. Second, we estimate the long-term cumulative value of each action $a_t$ to overcome the myopia of greedy optimization, and utilize these values to dynamically re-weight the importance of each timestep in policy updates. The specifics of this mechanism are detailed below.

**Instant Reward.** We formalize the generation process as a finite-time Markov Decision Process (MDP), characterized by the tuple $(\mathcal{S}, \mathcal{A}, \rho_0, P, R)$. At each reverse-time step $t \in \{T, ..., 1\}$, the model is in a latent state $s_t = x_t \in \mathcal{S}$ and takes an action $a_t \sim \pi(\cdot|s_t)$, which corresponds to the SDE exploration that transitions the state to $s_{t-1} = x_{t-1}$. A key challenge in this MDP is the absence of intermediate reward signals $R_t(s_t, a_t)$. This issue is exacerbated in diffusion models, where the heavy noise in early-stage images makes direct evaluation unreliable and semantically meaningless via a process reward model (Liang et al., 2024). To avoid this problem, we concep-

tualize the flow model as a one-step generation model. Specifically, at each sampling step $t$, after taking action $\boldsymbol{a}_t$ in the state $\boldsymbol{s}_t$ to reach $\boldsymbol{s}_{t-1}$, we perform a one-step deterministic ODE sampling from $\boldsymbol{s}_{t-1}$ to obtain a projected terminal state $\hat{\boldsymbol{x}}_0$. The instant reward $R_t(\boldsymbol{s}_t, \boldsymbol{a}_t)$ is defined as:

$$R_t(\boldsymbol{s}_t, \boldsymbol{a}_t) = \text{RM}(\hat{\boldsymbol{x}}_0, \boldsymbol{c}), \hat{\boldsymbol{x}}_0 = \boldsymbol{s}_{t-1} - (t-1)\boldsymbol{v}_\theta(\boldsymbol{s}_{t-1}, t-1), \boldsymbol{s}_{t-1} = f(\boldsymbol{s}_t, \boldsymbol{a}_t) \quad (9)$$

where $\boldsymbol{v}_\theta(\boldsymbol{s}_{t-1}, t-1)$ denotes velocity field predicted by the model, RM denotes reward model, $f$ denotes the SDE exploration (Eq. 7).

**Long-term Cumulative Value.** While the instant reward $R_t$ provides valuable per-step feedback, the policy that greedily optimizes for it would be myopic. Such a policy ignores the long-term consequences of an action, where a high immediate reward might steer the trajectory towards a sub-optimal future. To instill long-term foresight into the policy, we estimate the action value function $Q^\pi(\boldsymbol{s}_t, \boldsymbol{a}_t)$, which captures the expected cumulative discounted reward starting from action $\boldsymbol{a}_t$ in state $\boldsymbol{s}_t$ and subsequently following policy $\pi$, formulated as:

$$Q^\pi(\boldsymbol{s}_t, \boldsymbol{a}_t) = \mathbb{E}_\pi\left[\sum_{k=0}^{t-1}\gamma^k R_{t-k} \mid \boldsymbol{s}_t, \boldsymbol{a}_t\right] \quad (10)$$

where $\gamma \in [0, 1)$ is the discount factor. In practice, we leverages Monte-Carlo estimation (Sutton et al., 1998) over the sampling trajectory to assess the value $Q_t^i(\boldsymbol{s}_t, \boldsymbol{a}_t)$ of intermediate actions.

While the action value $Q_t^i(\boldsymbol{s}_t, \boldsymbol{a}_t)$ encapsulates the expected cumulative future reward, its absolute magnitude is discarded during standard advantage normalization. This transformation to a relative-only signal obscures the intrinsic value of each timestep, preventing the model from recognizing which actions contributed more significantly to the overall outcome. To recover this crucial information, we propose an explicit, value-driven weight $\omega_t^i$, which is designed to amplify the optimization signal for timesteps that lead to higher overall returns, dynamically assigning greater importance to more critical decisions within the generation process. It is formulated as:

$$\omega_t^i = \frac{Q_t^i(\boldsymbol{s}_t, \boldsymbol{a}_t)}{\text{mean}\left(\left\{Q_t^i(\boldsymbol{s}_t, \boldsymbol{a}_t)\right\}_{t=1}^T\right)} \quad (11)$$

### 3.3.2 Adaptive Dual Advantage estimation

The advantage function $A^\pi(\boldsymbol{s}_t, \boldsymbol{a}_t)$ quantifies the relative value of an action $\boldsymbol{a}_t$ compared to the expected policy behavior at state $\boldsymbol{s}_t$. To compute this advantage without the overhead of training a separate state value function $V^\pi(\boldsymbol{s}_t)$, GRPO instead employs a sample-based estimation. Specifically, it generates $G$ distinct trajectories from a single prompt $\boldsymbol{c}$ and computes the advantage relative to the mean of these sampled outcomes, formulated as:

$$\hat{A}_t^i(\boldsymbol{s}_t, \boldsymbol{a}_t) = \frac{Q_t^i(\boldsymbol{s}_t, \boldsymbol{a}_t) - \text{mean}\left(\left\{Q_t^i(\boldsymbol{s}_t, \boldsymbol{a}_t)\right\}_{i=1}^G\right)}{\text{std}\left(\left\{Q_t^i(\boldsymbol{s}_t, \boldsymbol{a}_t)\right\}_{i=1}^G\right)} \quad (12)$$

However, the standard GRPO advantage function $\hat{A}_t$ is a purely relative measure, which introduces critical flaws that destabilize optimization: (i) By applying identical optimization signals to sample groups of disparate absolute quality but similar relative structures, the advantage function stifles exploration for globally optimal strategies, effectively trapping the policy in local optima defined by relative gains. (ii) In low-variance stage, normalization by a near-zero standard deviation (eg. std=0.0008 in Fig. 2) forges an illusory advantage by amplifying trivial reward gaps, driving reward hacking over genuine quality improvements. (iii) During the late stages of policy convergence, the advantage signal collapses to zero as reward variance disappears, causing optimization to stagnate and risking policy collapse regardless of the samples' absolute quality. To address these flaws, we propose the Adaptive Dual Advantage Estimation (ADAE), formulated as:

$$\hat{A}_t^i(\boldsymbol{s}_t, \boldsymbol{a}_t) = \omega_t^i \cdot \frac{(1+\alpha) \cdot Q_t^i(\boldsymbol{s}_t, \boldsymbol{a}_t) - \text{mean}\left(\left\{Q_t^i(\boldsymbol{s}_t, \boldsymbol{a}_t)\right\}_{i=1}^G\right)}{\text{std}\left(\left\{Q_t^i(\boldsymbol{s}_t, \boldsymbol{a}_t)\right\}_{i=1}^G\right)} \quad (13)$$

where $\alpha$ is hyper-parameter, $\omega_t^i$ is value-driven weight for step re-weighting (Eq. 11). By adaptively merging the relative advantage dependent on reward diversity with robust global advantage, ADAE resolves the above flaws, achieving stable optimization and enabling higher-quality and more diverse generation. We present the complete VGPO training strategy in Algorithm 1.

---

**Algorithm 1** VGPO Training Process

---

**Input:** Reward model RM; Prompt dataset $\mathcal{C}$; Sampling steps $T$; Training steps $S$; Number of samples per prompt $G$; Temporal discount factor $\gamma$; Hyper-parameter $\alpha$

**Output:** Optimized model parameters $\theta$

1: Initial policy model $\pi_\theta$
2: **for** step $= 1, \cdots, S$ **do**
3:     Sample batch of prompts $C_b \sim \mathcal{C}$
4:     Update old policy model: $\pi_{\theta_{old}} \leftarrow \pi_\theta$
5:     **for** each prompt $c \in C_b$ **do**
6:         Init the noise $\boldsymbol{x}_T \sim \mathcal{N}(0, \mathbf{I})$
7:         **for** generate $i$-th image from $i = 1$ **to** $G$ **do**
8:             **for** sampling timestep $t = T$ **to** 1 **do**
9:                 Use SDE Sampling to get $\boldsymbol{x}_{t-1}^i \leftarrow$ Eq. 7
10:                Use One-Step ODE Sampling to get $\hat{\boldsymbol{x}}_0^i$ and instant reward $R_t^i \leftarrow$ Eq. 9
11:             **end for**
12:             Calculate long-term value $Q_t^i \leftarrow$ Eq. 10
13:             Calculate value-driven weight $\omega_t^i \leftarrow$ Eq. 11
14:             Calculate adaptive dual advantage: $\hat{A}_t^i \leftarrow$ Eq. 13
15:         **end for**
16:     **end for**
17:     Update policy model via gradient ascent: $\theta \leftarrow \theta + \eta \nabla_\theta \mathcal{J}$
18: **end for**

---

## 4 EXPERIMENTS

### 4.1 EXPERIMENTAL SETUP

Following Flow-GRPO, we evaluate our method on three distinct tasks: compositional image generation in GenEval (Ghosh et al., 2023) , visual text rendering (Chen et al., 2023) in OCR (Gong et al., 2025) and human preference alignment in PickScore (Kirstain et al., 2023). For all tasks, the objective is to maximize the reward score while preserving overall image quality. We adopt SD-3.5 (Esser et al., 2024) as the base model, consistent with the baseline. To demonstrate that our method effectively mitigates reward hacking, we assess performance from two fronts: (i) Task-specific accuracy on in-distribution test sets. (ii) General image quality on DrawBench (Saharia et al., 2022). The latter is measured by a suite of metrics encompassing image quality (Aesthetic (Schuhmann, 2022), DeQA (You et al., 2025)) and preference score (ImageReward (Xu et al., 2023), PickScore (Kirstain et al., 2023), UnifiedReward (Wang et al., 2025b)), ensuring that improvements in task alignment do not compromise generative quality. See Appendix A for details.

### 4.2 MAIN RESULTS

**Quantitative Analysis.** Our quantitative evaluation, detailed in Tab. 1, confirms the comprehensive superiority of VGPO across three benchmarks. In the absence of KL regularization (w/o KL), VGPO not only achieves steady improvements in task-specific metrics, but it enhances general image quality and preference score, significantly mitigating the reward hacking issue compared to Flow-GRPO. For example, in the compositional generation task without KL regularization (w/o KL), VGPO improves the GenEval score by 0.02 (from 0.95 to 0.97) while simultaneously boosting the average score across five quality and preference metrics by 9%. This pattern of dual improvement is also evident in visual text rendering, where OCR accuracy rises from 0.93 to 0.95 alongside substantial enhancements in image quality. Furthermore,

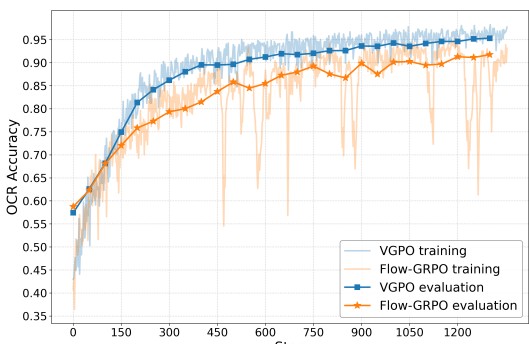

Figure 3: **Learning Curves** with KL on OCR.

Table 1: **Comparison Results** on Compositional Image Generation, Visual Text Rendering, and Human Preference Alignment benchmarks, evaluated by task performance, image quality, and preference score. OCR: OCR Accuracy; ImgRwd: ImageReward; UniRwd: UnifiedReward.

| Model | Task Metric | | | Image Quality | | Preference Score | | |
|---|---|---|---|---|---|---|---|---|
| | GenEval↑ | OCR↑ | PickScore↑ | Aesthetic↑ | DeQA↑ | ImgRwd↑ | PickScore↑ | UniRwd↑ |
| SD3.5-M | 0.63 | 0.59 | 21.72 | 5.39 | 4.07 | 0.87 | 22.34 | 3.07 |
| | | | *Compositional Image Generation* | | | | | |
| Flow-GRPO (w/o KL) | 0.95 | - | - | 4.93 | 2.77 | 0.44 | 21.16 | 2.59 |
| VGPO (w/o KL) | 0.97 (+0.02) | - | - | 5.23 (+0.3) | 3.45 (+0.68) | 0.94 (+0.50) | 22.00 (+0.84) | 3.00 (+0.41) |
| Flow-GRPO (w/ KL) | 0.95 | - | - | 5.25 | 4.01 | 1.03 | 22.37 | 3.18 |
| VGPO (w/ KL) | 0.96 (+0.01) | - | - | 5.41 (+0.16) | 4.05 (+0.04) | 1.09 (+0.06) | 22.59 (+0.22) | 3.23 (+0.05) |
| | | | *Visual Text Rendering* | | | | | |
| Flow-GRPO (w/o KL) | - | 0.93 | - | 5.13 | 3.66 | 0.58 | 21.79 | 2.82 |
| VGPO (w/o KL) | - | 0.95 (+0.02) | - | 5.33 (+0.2) | 3.98 (+0.32) | 0.90 (+0.32) | 22.17 (+0.38) | 3.07 (+0.25) |
| Flow-GRPO (w/ KL) | - | 0.92 | - | 5.32 | 4.06 | 0.95 | 22.44 | 3.14 |
| VGPO (w/ KL) | - | 0.94 (+0.02) | - | 5.34 (+0.02) | 4.08 (+0.02) | 0.98 (+0.03) | 22.44 (+0.0) | 3.14 (+0.0) |
| | | | *Human Preference Alignment* | | | | | |
| Flow-GRPO (w/o KL) | - | - | 23.41 | 6.15 | 4.16 | 1.24 | 23.56 | 3.33 |
| VGPO (w/o KL) | - | - | 23.55 (+0.14) | 5.97 (−0.18) | 4.18 (+0.02) | 1.28 (+0.04) | 23.70 (+0.14) | 3.34 (+0.01) |
| Flow-GRPO (w/ KL) | - | - | 23.31 | 5.92 | 4.22 | 1.28 | 23.53 | 3.38 |
| VGPO (w/ KL) | - | - | 23.41 (+0.10) | 5.90 (−0.02) | 4.23 (+0.01) | 1.32 (+0.04) | 23.61 (+0.08) | 3.39 (+0.01) |

this superiority persists under KL regularization. As depicted in Fig. 3, in addition to accelerated convergence(only 650 training steps to match the peak performance of Flow-GRPO), VGPO (w/ KL) exhibits markedly improved training stability, culminating in a higher final accuracy.

**Qualitative Analysis.** The quantitative findings are further corroborated by our qualitative analysis, with representative visualizations presented in Fig. 4. For the visual text rendering task, the first and fourth columns highlight VGPO's superior text accuracy. Notably, the second column reveals that VGPO maintains strong visual diversity even after successfully rendering text, effectively resisting the tendency to overfit the reward and collapse into a single stylistic mode. In the human preference alignment task, examples in the third and fifth columns showcase VGPO's exceptional capability in rendering fine-grained details and complex textures, producing images with heightened realism and visual fidelity. See Appendix C for per-category performance on the GenEval benchmark, and Appendix F for more visualizations.

## 4.3 ABLATION ANALYSIS

We conducted ablation studies to investigate the individual contributions of our two core components: TCRM and ADAE. Using the OCR task (w/o KL) as a case study, Tab. 2 shows that both components independently improve task accuracy and enhance overall image quality. TCRM's primary contribution is accelerating convergence, as shown in

Table 2: **Ablation Study** of main components.

| TCRM | ADAE | OCR | Aes | DeQA | ImgRwd | PickScore | UniRwd |
|---|---|---|---|---|---|---|---|
| | | 0.93 | 5.13 | 3.66 | 0.58 | 21.79 | 2.82 |
| ✓ | | 0.94 | 5.10 | 3.86 | 0.73 | 21.98 | 2.92 |
| | ✓ | 0.94 | 5.21 | 3.88 | 0.86 | 22.27 | 3.02 |
| ✓ | ✓ | 0.95 | 5.33 | 3.98 | 0.90 | 22.17 | 3.07 |

Fig. 5(a)(b). By providing a dense and granular optimization signal at each step, TCRM guides the model more efficiently towards the optimal solution, thereby reducing the sample inefficiency associated with sparse rewards. This allows the model to reach target performance in fewer training steps. In contrast, ADAE ensures training stability. It provides a robust optimization signal that prevents policy stagnation, and this sustained learning gradient translates directly into superior output quality. Fig. 5(c) confirms this, demonstrating that introducing ADAE leads to significant improvements in image quality and preference scores while maintaining an equivalent level of task performance. Crucially, this holistic enhancement is achieved without sacrificing visual quality, confirming that ADAE effectively mitigates reward hacking rather than narrowly overfitting to the reward signal.

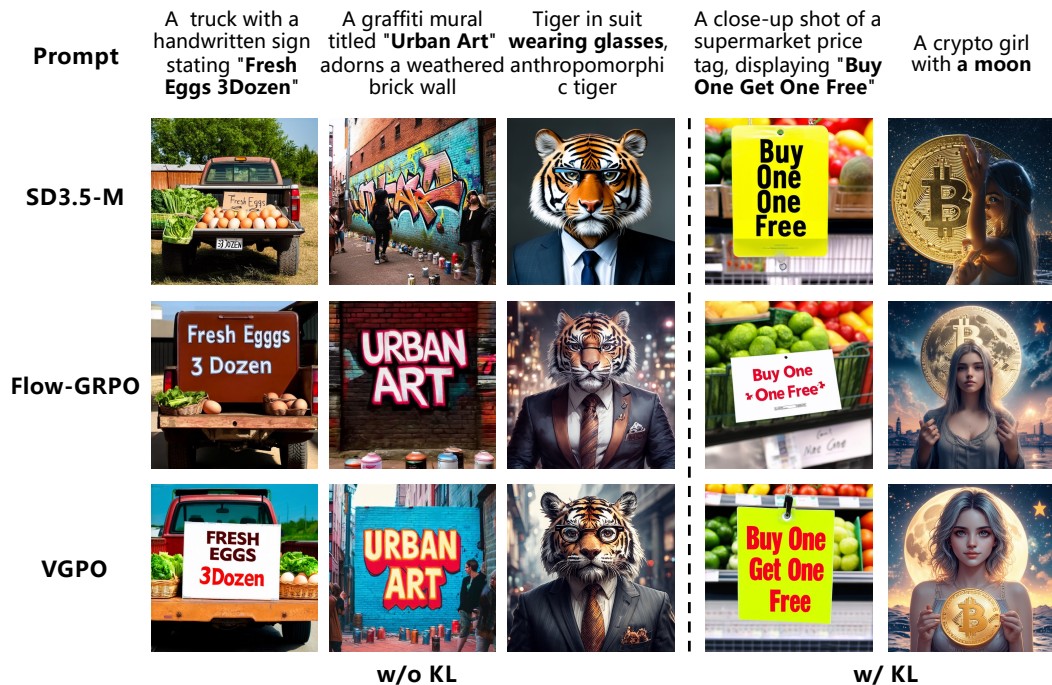

Figure 4: **Qualitative Comparison.** VGPO achieves superior performance in task accuracy, image quality and fine-grained details.

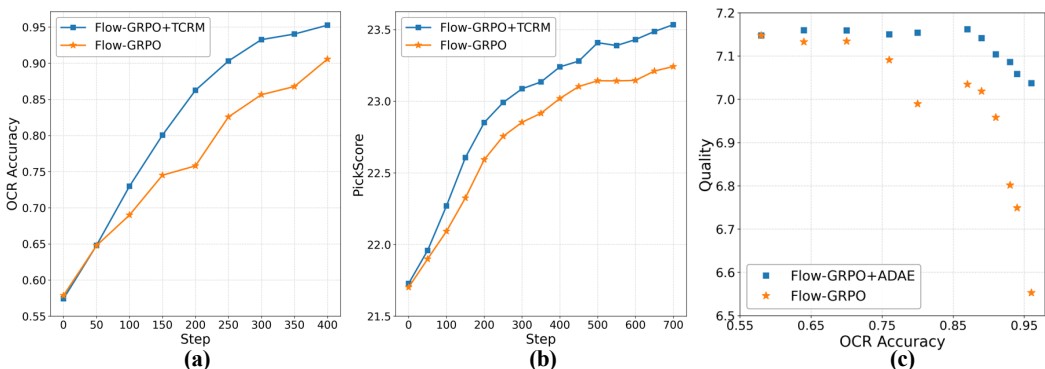

Figure 5: **Ablation Analysis.** The impact of TCRM is evaluated on the (a) OCR and (b) PickScore benchmarks, while (c) assesses ADAE's contribution to image quality at an equivalent OCR accuracy. Quality is the average of the five metrics across the "Image Quality" and "Preference Score".

## 5 CONCLUSION

In this paper, we observe that directly applying GRPO frameworks to flow matching models introduces two critical limitations: a misalignment between the exploration process and the final reward outcome, caused by the uniform application of sparse terminal rewards, and reliance on reward diversity renders it vulnerable to optimization stagnation as this diversity decreases. To address these problems, we propose Value-Anchored Group Policy Optimization (VGPO). At its core, VGPO facilitates granular credit assignment by transforming sparse terminal rewards into dense, forward-looking process values. Concurrently, it incorporates absolute values into the advantage computation to maintain a persistent optimization signal. Extensive experiments on three benchmarks demonstrate that VGPO achieves significant improvements in both task-specific accuracy and general image quality, while effectively mitigating reward hacking.

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

# A    DETAILS ON THE EXPERIMENTAL SETUP

## A.1    DATASET

For the compositional image generation task, we employ the GenEval (Ghosh et al., 2023) benchmark to assess performance on complex compositional prompts. Training set of $50k$ prompts are constructed using the official GenEval scripts and test set of $2k$ prompts is strictly deduplicated. The GenEval dataset includes six tasks, for which the prompt sampling ratio is set to Position : Counting : Attribute Binding : Colors : Two Objects : Single Object = $7:5:3:1:1:0$. For the visual text rendering task, we utilize a dataset of $20k$ training and $1k$ test prompts, with an OCR model (Gong et al., 2025) serving as the reward model. For the human preference alignment, the objective is to align T2I model with human aesthetics using a dataset of $25k$ training and $2k$ test prompts, where PickScore (Kirstain et al., 2023) as the reward model.

## A.2    HYPERPARAMETER SETTINGS

For each optimization step, we set group size $G = 24$, batch size to 6, each epoch consists of 8 batches and performs two gradient updates. We use a sampling timestep $T = 10$ and an evaluation timestep $T = 40$ to generate images with a resolution of $512$. Other settings include a noise level $a = 0.7$, a temporal discount factor $\gamma = 0.9$, hyper-parameter $\alpha = 0.1 * std$ for GenEval and OCR and $\alpha = 0.01 * std$ for PickScore. The KL ratio $\beta$ is set to $0.04$ for GenEval and Text Rendering, and $0.01$ for PickScore. In particular, to accelerate convergence, the term $\alpha$ is applied only during the first five sampling steps.

## A.3    QUALITY METRICS

The details of task accuracy metrics and quality metrics are as follows:

- GenEval (Ghosh et al., 2023): This metric assesses T2I models on complex compositional prompts across six difficult compositional image generation tasks. Its official evaluation pipeline detects object bounding boxes and colors, then infers their spatial relations. Rewards are rule-based: (i) Counting: $r = 1 - |N_{gen} - N_{ref}| / N_{ref}$; (ii) Position / Color: If the object count is correct, a partial reward is assigned; the remainder is granted when the predicted position or color is also correct.

- OCR accuracy (Gong et al., 2025): This metric quantifies the character-level correctness of the rendered text. It is calculated with the reward $r = \max(1 - N_e/N_{ref}, 0)$, where $N_e$ is the minimum edit distance between the rendered text and the target text and $N_{ref}$ is the number of characters inside the quotation marks in the prompt.

- PickScore (Kirstain et al., 2023): The PickScore model is obtained by fine-tuning a CLIP model(Radford et al., 2021) on Pick-a-Pic, a large-scale human preference dataset that records real users' choices between two images generated from the same prompt. It provides a comprehensive score highly consistent with human judgment to evaluate the overall quality of generated images, which primarily includes text-image alignment and visual aesthetic quality.

- Aesthetic score (Schuhmann, 2022): a CLIP-based linear regressor that predicts an image's aesthetic score.

- DeQA score (You et al., 2025): a multimodal large language model-based image quality assessment (IQA) model that quantifies an overall perceived quality, determined by factors like distortions, texture damage, and AI artifacts, by modeling the score distribution as a soft label and leveraging a fidelity loss.

- ImageReward (Xu et al., 2023): the first general-purpose text-to-image human preference reward model, designed to capture key human preference dimensions including text-image alignment, image fidelity, and harmlessness.

- UnifiedReward (Wang et al., 2025b): the first unified reward model for both multimodal understanding and generation assessment, developed on a large-scale human preference dataset spanning image and video tasks. By jointly learning to assess these diverse visual tasks, it demonstrates a significant synergistic effect, achieving substantial performance improvements over existing specialized reward models on multiple evaluation benchmarks.

## B   THEORETICAL ANALYSIS

This section provides the formal mathematical proof for the claim that ADAE automatically switches to optimizing absolute metrics when reward diversity is entirely depleted.

**Preliminaries and Definitions.** First, we recall the relevant equations. The Adaptive Dual Advantage Estimation (ADAE) is defined as:

$$\hat{A}_t^i \left( \boldsymbol{s}_t, \boldsymbol{a}_t \right) = \omega_t^i \cdot \frac{(1 + \alpha) \cdot Q_t^i \left( \boldsymbol{s}_t, \boldsymbol{a}_t \right) - \text{mean} \left( \left\{ Q_t^i \left( \boldsymbol{s}_t, \boldsymbol{a}_t \right) \right\}_{i=1}^G \right)}{\text{std} \left( \left\{ Q_t^i \left( \boldsymbol{s}_t, \boldsymbol{a}_t \right) \right\}_{i=1}^G \right)} \tag{14}$$

where $G$ is the group size. Crucially, as stated in Appendix A.2, the hyperparameter $\alpha$ is not a constant but is defined as a function of the group's reward standard deviation:

$$\alpha = k \cdot \text{std} \left( \left\{ Q_t^i \left( \boldsymbol{s}_t, \boldsymbol{a}_t \right) \right\}_{i=1}^G \right) \tag{15}$$

where $k$ is a small, constant hyperparameter (e.g., $k = 0.1$ for GenEval and OCR, $k = 0.01$ for PickScore). For notational simplicity within this proof, we fix the timestep $t$ and state $\boldsymbol{s}_t$, and denote $Q_i = Q_t^i \left( \boldsymbol{s}_t, \boldsymbol{a}_t \right), \mu = \text{mean} \left( \{Q_i\}_{i=1}^G \right), \sigma = \text{std} \left( \{Q_i\}_{i=1}^G \right)$

The ADAE formula can then be rewritten as follows:

$$\hat{A}_t^i = \omega_i \cdot \frac{(1 + \alpha) \cdot Q_i - \mu}{\sigma} \tag{16}$$

**Limiting Condition.** We analyze the behavior of ADAE under the condition that "reward diversity is entirely depleted." Mathematically, this corresponds to the scenario where the standard deviation of the action values within the group approaches zero $\sigma \to 0$. It implies that all individual action values $Q_i$ in the group are converging to a single, common value, which is their mean $C$.

$$\lim_{\sigma \to 0} Q_i = C \quad \text{for all} \quad i \in \{1, \cdots, G\} \tag{17}$$

Consequently, the mean also converges to $C$:

$$\lim_{\sigma \to 0} \mu = C \tag{18}$$

**Derivation of the Limit.** As $\sigma \to 0$, both $Q_i$ and $\mu$ approach the constant $C$. Using this property, the limit can be resolved through the following chain of equalities:

$$\begin{aligned} \lim_{\sigma \to 0} \hat{A}_i &= \lim_{\sigma \to 0} \left( \omega_i \cdot \frac{(1 + k \cdot \sigma) \cdot Q_i - \mu}{\sigma} \right) \\ &= \lim_{\sigma \to 0} \left( \omega_i \cdot \frac{(1 + k \cdot \sigma) \cdot C - C}{\sigma} \right) \\ &= \omega_i k \cdot C \end{aligned} \tag{19}$$

This final result shows that as reward diversity vanishes, the advantage signal $\hat{A}_i$ converges to a non-zero value, thus proving the automatic switching behavior of ADAE.

In the limit where reward diversity completely vanishes, unlike the standard GRPO signal which collapses to zero, the ADAE advantage signal converges to a stable, non-zero value. This provides a persistent optimization gradient for the policy, proving that it ultimately turns to optimizing absolute metrics. In addition, in standard GRPO, a small $\sigma$ disproportionately amplifies meaningless, tiny differences between $Q_i$ values, leading to large and unstable advantages. In ADAE, the presence of the $\alpha$ term acts as a "stabilizer." It balances the noise amplification effect caused by dividing by a small $\sigma$, making the advantage signal more robust and thus suppressing overfitting to minor reward fluctuations.

## C   MORE EXPERIMENTS RESULTS IN GENEVAL

Tab. 3 details the performance of our VGPO across each subtask, achieving an overall score of 0.96 in the GenEval evaluation.

Table 3: **GenEval Result.** Best scores are highlighted in blue, second-best in green. Results for models are from Flow-GRPO. Obj: Object; Attr: Attribution.

| Model | Overall | Single Obj. | Two Obj. | Counting | Colors | Position | Attr. Binding |
|---|---|---|---|---|---|---|---|
| *Diffusion Models* | | | | | | | |
| LDM (Rombach et al., 2022) | 0.37 | 0.92 | 0.29 | 0.23 | 0.70 | 0.02 | 0.05 |
| SD1.5 (Rombach et al., 2022) | 0.43 | 0.97 | 0.38 | 0.35 | 0.76 | 0.04 | 0.06 |
| SD2.1 (Rombach et al., 2022) | 0.50 | 0.98 | 0.51 | 0.44 | 0.85 | 0.07 | 0.17 |
| SD-XL (Podell et al., 2023) | 0.55 | 0.98 | 0.74 | 0.39 | 0.85 | 0.15 | 0.23 |
| DALLE-2 (Ramesh et al., 2022) | 0.52 | 0.94 | 0.66 | 0.49 | 0.77 | 0.10 | 0.19 |
| DALLE-3 (Betker et al., 2023) | 0.67 | 0.96 | 0.87 | 0.47 | 0.83 | 0.43 | 0.45 |
| *Autoregressive Models* | | | | | | | |
| Show-o (Xie et al., 2024) | 0.53 | 0.95 | 0.52 | 0.49 | 0.82 | 0.11 | 0.28 |
| Emu3-Gen (Wang et al., 2024) | 0.54 | 0.98 | 0.71 | 0.34 | 0.81 | 0.17 | 0.21 |
| JanusFlow (Ma et al., 2025) | 0.63 | 0.97 | 0.59 | 0.45 | 0.83 | 0.53 | 0.42 |
| Janus-Pro-7B (Chen et al., 2025b) | 0.80 | 0.99 | 0.89 | 0.59 | 0.90 | 0.79 | 0.66 |
| GPT-4o (Hurst et al., 2024) | 0.84 | 0.99 | 0.92 | 0.85 | 0.92 | 0.75 | 0.61 |
| *Flow Matching Models* | | | | | | | |
| FLUX.1 Dev (Labs, 2024) | 0.66 | 0.98 | 0.81 | 0.74 | 0.79 | 0.22 | 0.45 |
| SD3.5-L (Esser et al., 2024) | 0.71 | 0.98 | 0.89 | 0.73 | 0.83 | 0.34 | 0.47 |
| SANA-1.5 4.8B (Xie et al., 2025) | 0.81 | 0.99 | 0.93 | 0.86 | 0.84 | 0.59 | 0.65 |
| SD3.5-M (Esser et al., 2024) | 0.63 | 0.98 | 0.78 | 0.50 | 0.81 | 0.24 | 0.52 |
| Flow-GRPO (Liu et al., 2025a) | 0.95 | 1.00 | 0.99 | 0.95 | 0.92 | 0.99 | 0.86 |
| **VGPO** | 0.96 | 1.00 | 0.99 | 0.98 | 0.96 | 0.97 | 0.88 |

## D LIMITATION AND FUTURE WORK

Although our method demonstrates significant improvements, its validation remains limited to a single reward setting within the T2I domain. In the future, we plan to integrate signals from multiple reward models to achieve more comprehensive alignment with complex human preferences. Additionally, extending our framework to T2V generation is a crucial next step. Effective video alignment requires optimization across complex criteria such as realism, temporal smoothness, and physical plausibility. A major bottleneck in this domain is the scarcity of high-quality video reward models (whose development speed lags behind foundational video generation models), making the training of video generation reward models equally critical.

## E USE OF LARGE LANGUAGE MODELS (LLMS)

In accordance with ICLR 2026 policy, we report the use of a Large Language Model (LLM) during the preparation of this manuscript. Its application was strictly limited to enhancing the linguistic quality of the text, including improving sentence structure and ensuring grammatical correctness. The entirety of the scientific contributions in this paper, spanning problem formulation, method design, experimental validation, and final conclusions, is attributable solely to the authors. The final manuscript has been critically reviewed by all authors, who collectively assume full responsibility for its accuracy and integrity.

## F   MORE VISUALIZED RESULTS

| **SD3.5-M** | **Flow-GRPO w/o KL** | **VGPO** |
|:---:|:---:|:---:|

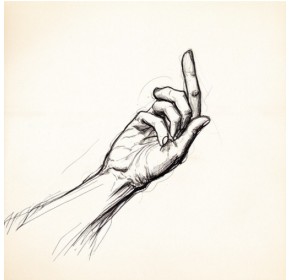 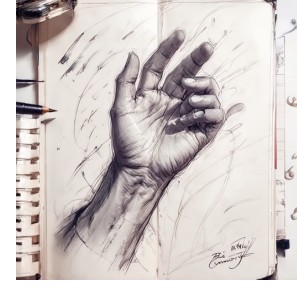 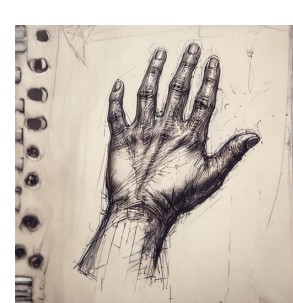

A detailed sketch of a left hand

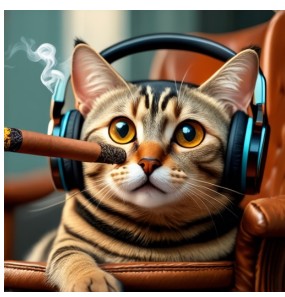 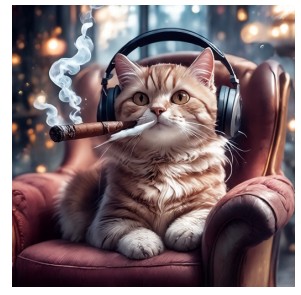 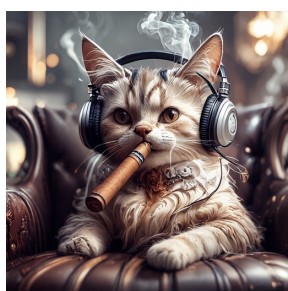

A cat smoking a cigar and wearing headphones, lying on a chair, 4k, 3d carton style

**w/ KL**

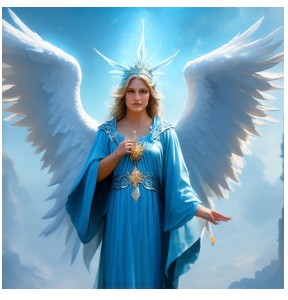 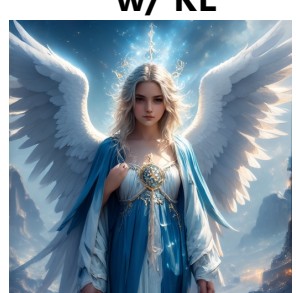 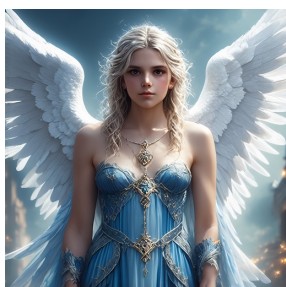

An epic angel dressed in blue with white wings

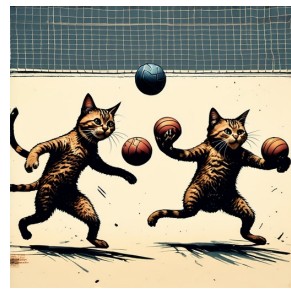 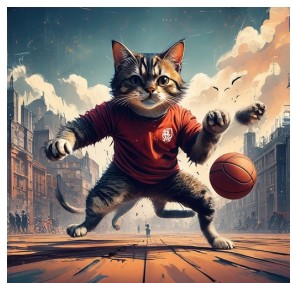 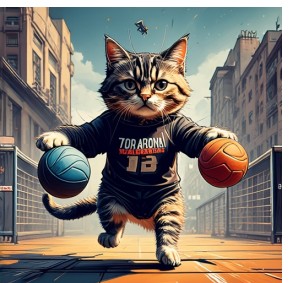

Anthropomorphic Cats playing dodgeball, by dan mumford and Banksy

Figure 6: Comparison of the visualization results between the SD3.5-M, Flow-GRPO and VGPO trained with **PickScore** reward.

| **SD3.5-M** | **Flow-GRPO** | **VGPO** |
| --- | --- | --- |
| | **w/o KL** | |

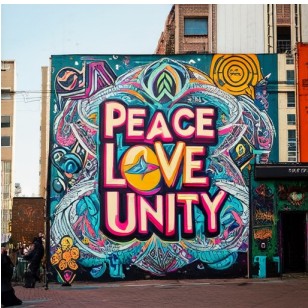 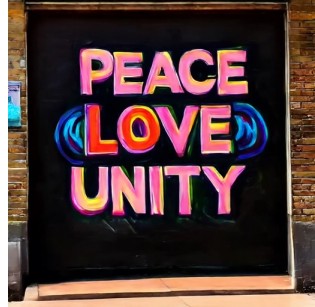 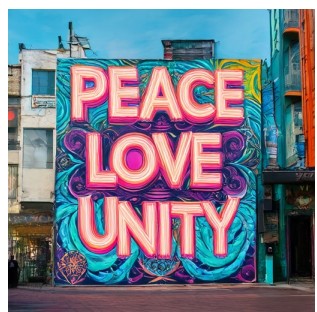

A vibrant street art mural , featuring the words "**Peace Love Unity**"

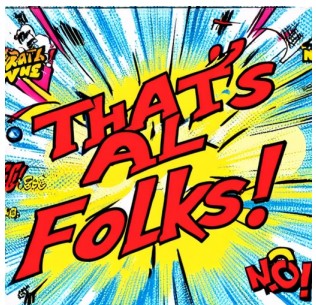 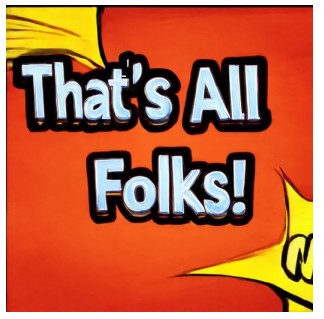 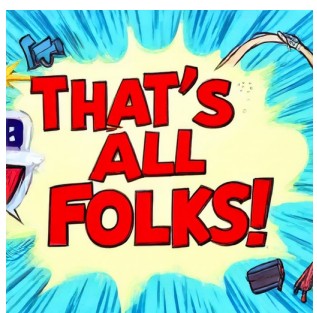

A comic strip panel with a colorful background, featuring text that exclaims
"**That's All Folks!**"

**w/ KL**

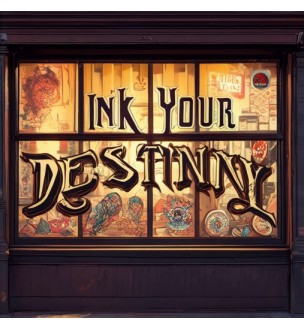 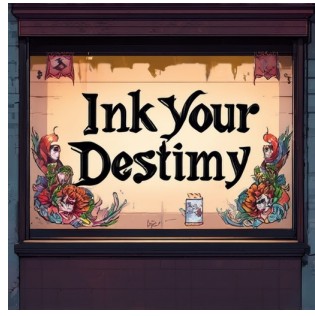 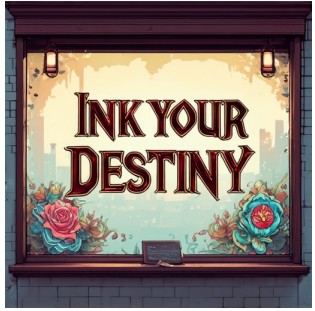

A tattoo parlor window with "**Ink Your Destiny**" in Gothic letters

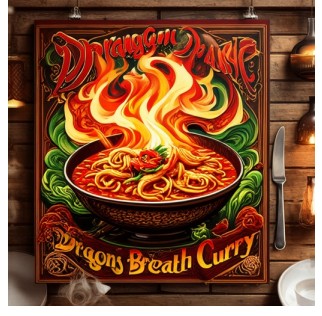 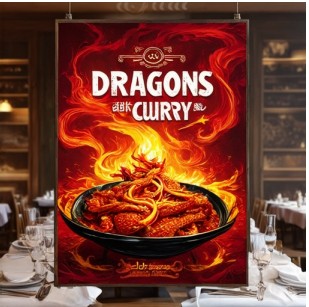 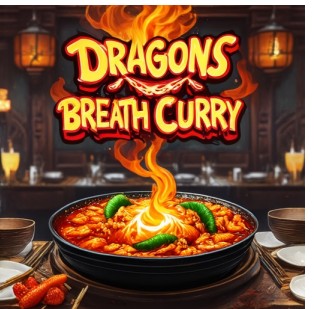

A restaurant menu with a detailed illustration of "**Dragons Breath Curry**",
steaming hot with a swirl of spicy smoke

Figure 7: Comparison of the visualization results between the SD3.5-M, Flow-GRPO and VGPO trained with **OCR** reward.

| **SD3.5-M** | **Flow-GRPO** **w/o KL** | **VGPO** |
|:---:|:---:|:---:|
| 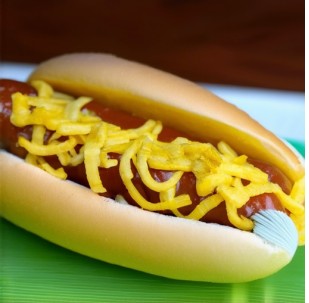 | 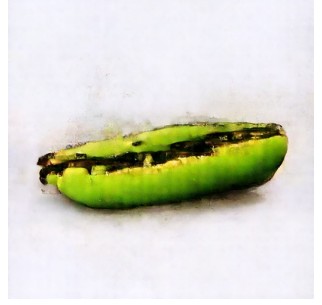 | 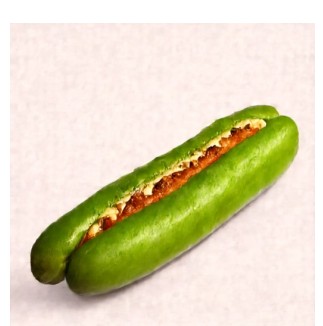 |

A photo of a green hot dog

| | | |
|:---:|:---:|:---:|
| 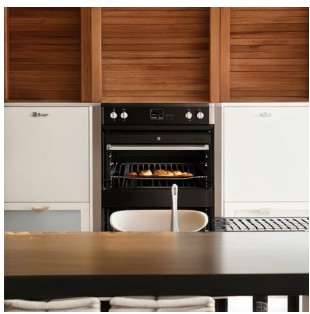 | 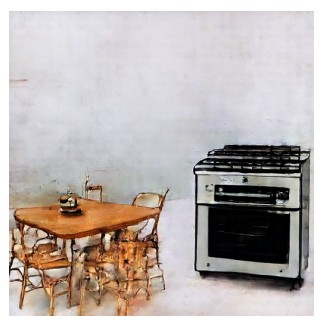 | 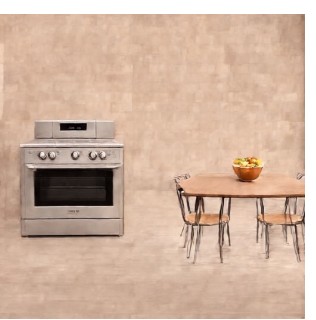 |

A photo of a dining table right of an oven

**w/ KL**

| | | |
|:---:|:---:|:---:|
| 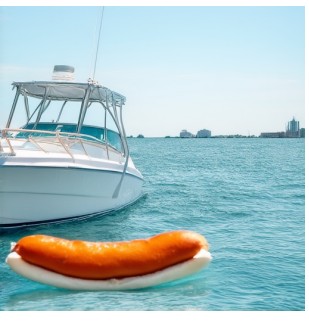 | 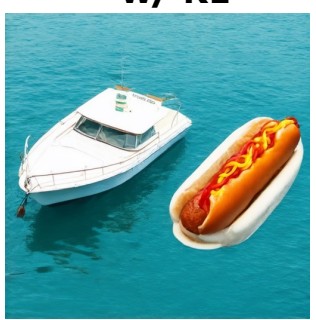 | 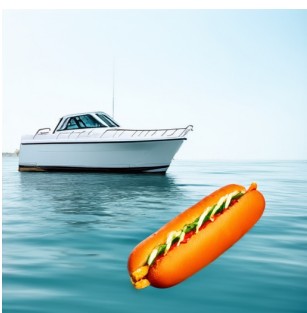 |

A photo of a white boat and an orange hot dog

| | | |
|:---:|:---:|:---:|
| 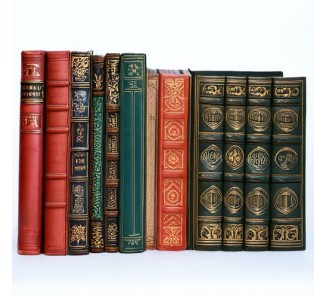 | 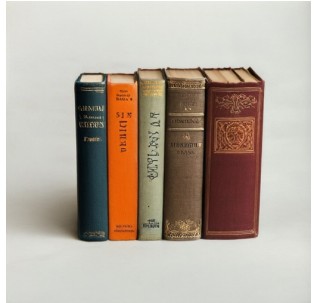 | 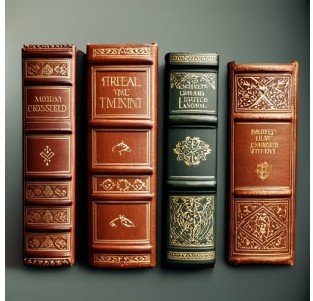 |

A photo of four books

Figure 8: Comparison of the visualization results between the SD3.5-M, Flow-GRPO and VGPO trained with **GenEval** reward.

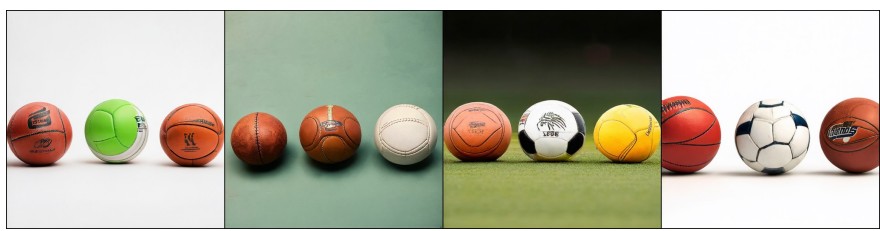

Single object: a photo of a teddy bear

Two object: a photo of a tennis racket and a bicycle

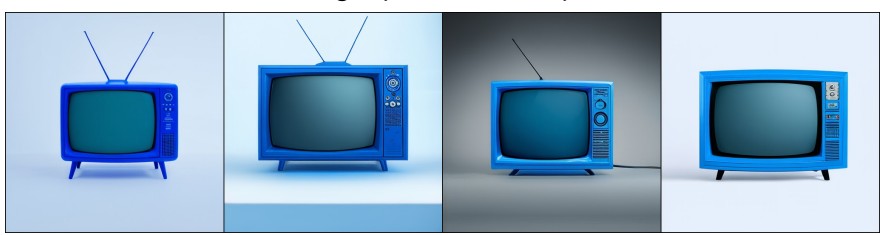

Counting: a photo of three sports balls

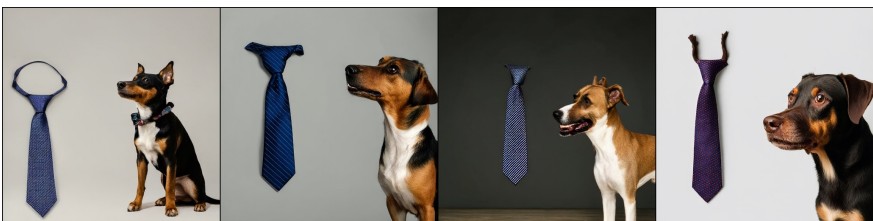

Colors: a photo of a blue tv

Position: a photo of a dog right of a tie

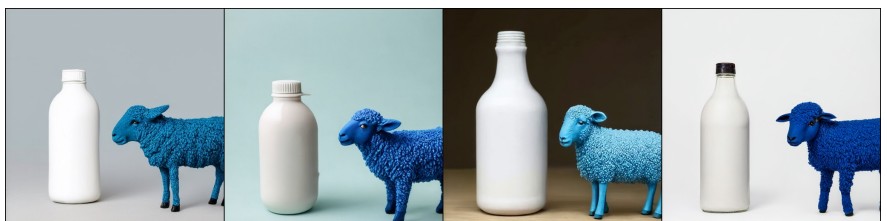

Attribute binding: a photo of a white bottle and a blue sheep

Figure 9: More visualization results of VGPO on **GenEval** benchmark.

