# OpenReview forum: "Value-Anchored Group Policy Optimization for Flow Models"
_ICLR.cc/2026/Conference — ICLR 2026 Conference Withdrawn Submission_

### Official Review · Reviewer_knyz · 2025-10-25

**Soundness:** 1
**Presentation:** 3
**Contribution:** 3
**Rating:** 4
**Confidence:** 4

**Summary:**

This paper identifies and addresses two major limitations when applying GRPO to flow-based generative models: poor temporal credit assignment and optimization stagnation. The authors introduce value-anchored group policy optimization (VGPO) to address these issues by leveraging a temporal cumulative reward mechanism and advantage estimation. The authors demonstrate, through experiments across three benchmarks (compositional generation, text rendering, and human preference), that VGPO achieves state-of-the-art image quality.

**Strengths:**

- The paper identifies a fundamental mismatch when applying GRPO to flow-based generation, pinpointing the two critical limitations of poor temporal credit assignment and reliance on reward diversity. To the best of my knowledge, limited work has explored this problem before.
- The authors' delivery of the proposed approach is clear and well-supported by the issues raised. The VGPO approach is easy to follow and intuitively makes sense.
- Code is provided for reproducibility.
- Comprehensive experiments demonstrate the superior performance of the proposed approach compared to the vanilla GRPO models.

**Weaknesses:**

- My most significant concern involves the validity of using one-step extrapolation in the temporal cumulative reward mechanism. Flow matching models learn the **marginal** vector field at each noisy data point. Therefore, it is not guaranteed to be a straight sampling path and requires ODE/SDE solvers with normally tens to hundreds of sampling steps for decent generation results. This is also the reason why techniques like rectified flow and consistency models emerge for few-step generation while maintaining the marginals. Therefore, the simple one-step extrapolation from the early sampling stage will likely deviate from the distribution of the learned data, leading to less credible rewards. It remains unclear, at least mathematically, how such stepwise rewards from a shifted distribution can still guarantee the convergence of the GRPO/DPO-based methods.
- As the proposed method requires the calculation of rewards on each intermediate noisy sample in the iterative inference process, the practical running time of the algorithm may be larger compared to the vanilla GRPO/DPO approaches, especially for rewards that rely on other ML models, such as CLIP score.
- Baselines compared in the paper remain limited. Only the vanilla GRPO was compared. DPO-based approaches, which share a similar spirit, should also be considered.

**Questions:**

- Can you derive or discuss the validity of using the one-step extrapolation even when the marginal probability path is not guaranteed to be straight?
- Can you provide detailed running time comparisons between the proposed approach and the baselines?
- How is the performance of DPO-based models on this task?

---

### Official Review · Reviewer_66RW · 2025-10-30

**Soundness:** 2
**Presentation:** 3
**Contribution:** 2
**Rating:** 2
**Confidence:** 4

**Summary:**

The paper targets two issues when applying Group Relative Policy Optimization (GRPO) to flow‑matching text‑to‑image models: temporal miscredit from using a single terminal reward at every denoising step, and vanishing optimization signal when group rewards lose variance. There are two contributions methodological contributions: first, they added a Temporal Cumulative Reward Mechanism (TCRM) that converts the terminal reward into per‑step “instant” rewards by projecting each intermediate latent one ODE step to an approximate and computing a reward on it, then forms discounted action values $Q_t$ ; and second, Adaptive Dual Advantage Estimation (ADAE) that mixes a relative term with an absolute‑value term so that the advantage does not collapse when the group reward standard deviation goes to zero. The method is evaluated on GenEval, OCR text rendering, and PickScore alignment using SD‑3.5 as the base model. The headline results claim modest improvements over Flow‑GRPO on in‑distribution task metrics and on quality metrics, with the largest relative gains occurring when KL regularization is absent. Key equations and the training loop are clearly stated in Section 3 and Algorithm 1.

**Strengths:**

I think there is a strong motivation for this work as plainly appling GRPO to flow based models presents some issues, most prominently diversity of generated samples. I also like the simplicity of the propose methods, make it easy to implement and evaluate. The presentation of the paper is very clear as well, make it easy for readers to digest.

**Weaknesses:**

**Suprisingly lack of recent baselines for empirical comparision (even though it was mentioned in the work).** In related works section the authors clearly stated related works like Tempflow [1], Preflow [2], Mixgrpo [3], BranchGRPO [4]. In particular Tempflow also incorporate temporal reward signal and I think it should be incorporated into the empirical evaluations.

While I think an improvement with Flow-GRPO is great, **I believe some of the empirical gains are small and sometimes negative on quality**. There was a bit of overclaiming the effectiveness of the proposed method. An example of minimal gain is in Table 1 on page 8, with KL, GenEval improves from 0.95 to 0.96, OCR from 0.92 to 0.94, and PickScore from 23.31 to 23.41. Several aesthetic or DeQA scores are within 0.01 to 0.16. Without KL, the human‑alignment setting shows a drop in Aesthetic from 6.15 to 5.97. “Reward hacking mitigation” is asserted, yet some quality metrics degrade or move minimally. Quantifying the variance across seeds and reporting effect sizes would substantiate claims.

This method also results in **a non-trivial computational overhead** compared to Flow-GRPO. Flow‑GRPO evaluates the reward once per generated image, TCRM evaluates it $T$ times per image. A wall‑clock or FLOPs analysis is needed vs Flow-GRPO, and comparisons with efficiency‑focused variants like MixGRPO or BranchGRPO would be appropriate.

This is more tricky and I will classify it as minor weakness. The theory is limited to a limit case. Appendix B proves that ADAE does not vanish when $\sigma \to 0$. This is a desirable property, but it does not address finite‑$\sigma$, bias, stability, or KL‑regularized convergence.

[1] Xiaoxuan He, Siming Fu, Yuke Zhao, Wanli Li, Jian Yang, Dacheng Yin, Fengyun Rao, and
Bo Zhang. Tempflow-grpo: When timing matters for grpo in flow models, 2025.

[2] Yibin Wang, Zhimin Li, Yuhang Zang, Yujie Zhou, Jiazi Bu, Chunyu Wang, Qinglin Lu, Cheng Jin, and Jiaqi Wang. Pref-grpo: Pairwise preference reward-based grpo for stable text-to-image reinforcement learning, 2025a.

[3] Junzhe Li, Yutao Cui, Tao Huang, Yinping Ma, Chun Fan, Miles Yang, and Zhao Zhong. Mixgrpo:
Unlocking flow-based grpo efficiency with mixed ode-sde, 2025a.

[4] Yuming Li, Yikai Wang, Yuying Zhu, Zhongyu Zhao, Ming Lu, Qi She, and Shanghang Zhang.
Branchgrpo: Stable and efficient grpo with structured branching in diffusion models, 2025b.

**Questions:**

See weakness, mostly about missing baselines and empirical benchmarks.

1. What is the wall‑clock or GPU‑hour overhead of TCRM relative to Flow‑GRPO?
2. How sensitive is ADAE to reward‑scale changes across reward models?

---

### Official Review · Reviewer_82es · 2025-10-31

**Soundness:** 2
**Presentation:** 3
**Contribution:** 3
**Rating:** 4
**Confidence:** 3

**Summary:**

The paper proposes **Value-Anchored Group Policy Optimization (VGPO)** for online RL alignment of flow-matching image generators. It argues that directly applying GRPO to flows causes **(i)** faulty credit assignment by spreading a sparse terminal reward uniformly over timesteps and **(ii)** signal collapse when intra-group reward variance shrinks.

VGPO addresses this via two components: Temporal Cumulative Reward Mechanism (TCRM), which uses a one-step ODE projection to define stepwise “instant rewards” and accumulates them into action values $Q_t$, and Adaptive Dual Advantage Estimation (ADAE), which mixes relative (group-normalized) and absolute (value-anchored) advantages with timestep re-weighting $\omega_t$. Experiments on compositional generation (GenEval), OCR text rendering, and human-preference alignment (PickScore) show higher task scores and generally improved image-quality metrics versus Flow-GRPO, with learning curves indicating faster and smoother convergence when KL is used. (Method & motivation: Fig. 1–2; Eqs. (8)–(13); Alg. 1; Results: Tab. 1, Fig. 3–5.)

**Strengths:**

- **Problem diagnosis is specific and evidenced.** Eq. (8) exhibits the coupling of a time-dependent policy ratio with a time-independent terminal advantage, explaining why uniform credit across steps is misaligned for flows; Fig. 1 (left/right) visualizes sparse-vs-instant reward and the decline of group reward std during training.
- **Clean MDP formalization and sampling interface.** The paper casts reverse sampling as an MDP (Eq. (3)) and gives the SDE sampler used for policy exploration (Eq. (7)), making the injection of per-step RL signals well-defined.
- **TCRM turns terminal reward into process values with explicit formulas.** One-step ODE projection defines $R_t$ (Eq. (9)), discounted accumulation yields $Q_t$ (Eq. (10)), and timestep weights $\omega_t$ (Eq. (11)) prioritize impactful steps; implementation appears straightforward in **Algorithm 1**.
- **It is low in cost and simple in engineering.** TCRM only performs one more ODE projection in each inversion step (Equation (9)) without introducing additional PRM or critic; Algorithm 1 shows that it can be directly spliced with the existing Flow-GRPO training process (page 7).

**Weaknesses:**

- Inconsistent time scale/notation around Eq. (9). Eqs. (1)/(7) use continuous $t\in[0,1]$ while Algorithm 1 iterates discrete $t=T,\ldots,1$; Eq. (9) then uses $(t-1)$ both as an argument to v_\theta and as a scalar step, which conflicts with the $[0,1]$ convention when $T=10$ (Appendix A.2).
- “Monte-Carlo estimation” wording does not match the computation. TCRM’s instant reward comes from a deterministic one-step ODE projection (Eq. (9)), and $Q_t$ is summed along a single sampled trajectory (Eq. (10)); there is no explicit resampling of future randomness or variance estimate
- ADAE’s non-collapse guarantee hinges on $\alpha=k\,\mathrm{std}(Q)$ but this dependency is not stated in the main text. Eq. (13) treats $\alpha$ as a hyper-parameter; the limit proof (App. B, Eqs. (15)–(19)) requires $\alpha\propto\mathrm{std}$.
- Training schedule conflicts with the theory: $\alpha$ applied only in first five steps. Appendix A.2 notes $\alpha$ is enabled only for the first 5 sampling steps, which is inconsistent with the “$\sigma\to0$” limit argument that assumes the \alpha-rule always holds.
- Definition/stability of $\omega_t$ is underspecified. Eq. (11) divides by the mean over $t$ of $Q_t$; behavior is unclear if rewards can be negative or the mean is near zero outside the reported non-negative settings (Appendix A.3).
- Mismatch between training and evaluation steps (T=10 vs. 40) is not justified. The paper does not explain how $Q_t/\omega_t$ advantages transfer across different step counts (Appendix A.2).

**Questions:**

1. Is $t$ in Eqs. (1)/(7) continuous and in Algorithm 1/Eq. (9) discrete? If so, should Eq. (9) use $\tau_{t}=t/T$ and $\hat{x}0=x{t-1}-\tau_{t-1}v_\theta(x_{t-1},\tau_{t-x1})$? The current $(t-1)$ conflicts with $[0,1]$ when $T=10$. (Eqs. (1),(7),(9); Alg. 1; App. A.2.)
2. Does Eq. (9) integrate from $t-1$ to $0$ in one Euler step (“remaining time” as the step length)? Please provide the derivation and either an error bound or an empirical bias analysis versus the true terminal state. (Eq. (9).)
3. Is $Q_t$ in Eq. (10) computed from a single rollout or with resampling around the same $x_{t-1}$? If single-rollout, what does “Monte-Carlo” refer to, and is $\mathrm{Var}[Q_t]$ tracked? (Text near Eq. (10); Alg. 1.)
4. The limit result (App. B) assumes $\alpha=k\,\mathrm{std}$ (Eq. (15)), while App. A.2 enables $\alpha$ only in the first five steps. What is the formal definition and time schedule of $\alpha$ in the main method, and how sensitive are results to $k$? (Eq. (13); App. A.2/B.)
5. Domain and robustness of $\omega_t$. With Eq. (11), how is $\omega_t$ handled if $Q_t$ can be negative or the mean across time is near zero? What assumptions on reward sign/scale are required?
6. Train–test step mismatch. With training $T=10$ and evaluation $T=40$, how do the learned $\omega_t$/$\hat{A}_t$ translate across different discretizations? Is any renormalization used?

---

### Official Review · Reviewer_t96z · 2025-11-01

**Soundness:** 3
**Presentation:** 2
**Contribution:** 2
**Rating:** 4
**Confidence:** 4

**Summary:**

The paper proposes VGPO, an extension of GRPO for flow-matching image generation. It introduces TCRM, which defines per-step rewards via one-step ODE projection and estimates discounted Q-values, and ADAE, which refines group-normalized advantages using timestep weights and a reward variance term. Evaluations on GenEval, OCR, and PickScore show small to moderate gains over Flow-GRPO.

**Strengths:**

- Identifies real issues in applying GRPO to flow models (temporal misalignment and reward diversity collapse). I like this insight.

- Clear algorithmic description with simple implementation.

- Reasonable experimental coverage across three benchmarks.

**Weaknesses:**

- The novelty is limited. The “instant reward” idea and the way of generating dense per-step feedback are not new; Many papers have already utilized the one-step operator to obtain the outcome from a sample at noise level t, and then compute the reward for training or inference time scaling [1].

- The “long-term cumulative value” is simply a standard discounted Q-value estimation commonly used in reinforcement learning, given the instant reward.

- The design of the advantage function is heuristic and ad hoc. The motivation is clear, but why this paper’s proposal resolves these issues remains unclear in both empirical and theoretical aspects.

- Overall, the contributions are incremental adaptations of existing concepts with minor modifications. The justification for these changes is not well-motivated and well-supported. The introduction of ω and α (3.3.2) seems ad hoc, with no clear theoretical justification for why they should stabilize training or prevent reward collapse. The std is still on the denominator and can still collapse to 0. The explanation provided (based on the limit when reward std → 0) is algebraic and does not clarify the actual learning dynamics. No further analysis is provided to demonstrate how these modifications alter the gradient behavior or enhance convergence.

- The paper only shows standard training curves and small metric improvements.

- There are no experiments designed to directly test whether ADAE indeed mitigates the collapse or variance issues it claims to address. Ablations are shallow and do not isolate the real effect of omega or alpha.

- The proposed method appears computationally expensive because it requires one-step ODE projection and Monte Carlo Q estimation at every time step. The paper does not report the computation cost, reward-model calls, or wall-clock comparison with the baseline.


[1]  Ma, Nanye, et al. "Inference-time scaling for diffusion models beyond scaling denoising steps." arXiv preprint arXiv:2501.09732 (2025).

**Questions:**

1. How does the proposed “instant reward” approach fundamentally differ from prior methods that also derive intermediate or dense process rewards?

2. What is the real computational overhead compared to Flow-GRPO (in terms of reward evaluations, training time, and GPU cost)?

3. Why were omega = Q/mean(Q) and alpha = k * std chosen? Are these values empirically tuned or based on theoretical reasoning?

4. Can you show any experiment where reward diversity is intentionally reduced to demonstrate that ADAE truly prevents optimization collapse?

5. How noisy are the Monte Carlo Q estimates, and how many rollouts are required per step to stabilize them?

---

### Note · Authors · 2025-11-13

I have read and agree with the venue's withdrawal policy on behalf of myself and my co-authors.